# A register-based cohort study on the effectiveness and Safety of anti-PCSK9 treatment in persons with hyperlipidemia
Michael Asger Andersen [1] ✉, Anne Helms Andreasen[2], Lia Evi Bang[3], Espen Jimenez-Solem[1] & Tonny Studsgaard Petersen[1]

## Abstract

**Background** Dyslipidemia is a known risk factor for cardiovascular disease. While statins are the primary treatment, some individuals require additional lipid-lowering therapies, such as proprotein convertase subtilisin/kexin type 9 (*PCSK9*) inhibitors. Alirocumab and evolocumab have shown efficacy in reducing low-density lipoprotein cholesterol (LDL-C) levels and reduce the risk of major cardiovascular events (MACE) but have not been directly compared in clinical trials. This study aims to assess the effects of *PCSK9* inhibitors on LDL-C levels and evaluate the impact of a mandated switch from alirocumab to evolocumab.
**Methods** Taking advantage of the mandated switch in *PCSK9* treatment in Denmark, we conducted a register-based cohort study of 907 individuals with dyslipidemia treated with *PCSK9* inhibitors in the Capital Region of Denmark from 2016 to 2022. We analyzed LDL-C levels, treatment retention, and MACE, adjusting for variables such as age, sex, dose, and concurrent lipid-lowering medications.
**Results** We show that *PCSK9* inhibitors treatment resulted in a 49% reduction in LDL-C levels. Following a mandated switch from alirocumab to evolocumab, no significant difference was observed in LDL-C levels or adverse clinical outcomes, including MACE. Treatment discontinuation was most likely within the first 100 days, and no significant difference in discontinuation rates was found between the two drugs.
**Conclusions** Our study demonstrates that both alirocumab and evolocumab are effective in significantly reducing LDL-C levels in individuals with dyslipidemia. The mandated switch from alirocumab to evolocumab did not result in significant changes in LDL-C or clinical outcomes, suggesting that these treatments can be used interchangeably. These findings support the clinical equivalence of the two *PCSK9* inhibitors and may guide therapeutic decisions in lipid management.

## Plain Language Summary

Dyslipidemia is a condition where there are unhealthy levels of fats, i.e. cholesterol (LDL-C and HDL-C), in the blood, which can increase the risk of heart disease. To help manage this, certain drugs called PCSK9 inhibitors, like alirocumab and evolocumab, can be used to lower cholesterol levels. Here, we studied 907 people in Denmark who were treated with these drugs between 2016 and 2022. We found that these drugs reduced LDL-C by almost half. During the study, some people were required to switch from one drug (alirocumab) to another (evolocumab), and this switch did not lead to any changes in cholesterol levels or increase the risk of heart problems or death. Our findings suggest that both alirocumab and evolocumab are safe and effective options for lowering cholesterol, and they can be used interchangeably, which might help doctors make treatment decisions.

Low-density lipoprotein cholesterol (LDL-C) has been consistently linked to the development of atherosclerotic cardiovascular disease (ASCVD)[1]. Reducing LDL-C levels can inhibit the progression of atherosclerotic plaques and lower the risk of ASCVD[2]. It has been estimated that the risk of ASCVD can be reduced by as much as 22% over 5 years with each 1 mmol/L reduction in LDL-C[2]. Statins are widely accepted and effective treatments for dyslipidemia and have proven to effectively lower LDL-C levels in numerous randomized clinical trials[3].

However, some persons may require additional lipid-lowering medications even when taking high-dose statins[2,3].

Proprotein convertase subtilisin/kexin type 9 (*PCSK9*) plays a role in regulating LDL-C levels by binding to LDL receptors and inhibiting their recycling[4]. Inhibition of *PCSK9* can increase the recycling of LDL receptors, leading to a decrease in LDL-C in plasma[4]. Alirocumab and evolocumab are fully humanized monoclonal antibodies that bind plasma *PCSK9*, promoting its degradation. *PCSK9* antibodies (anti-PCSK9) are capable of

[1]Department of Clinical Pharmacology, Copenhagen University Hospital - Bispebjerg and Frederiksberg Hospital, Copenhagen, Denmark. [2]Center for Clinical Research and Prevention, Copenhagen University Hospital - Bispebjerg and Frederiksberg, Copenhagen, Denmark. [3]Department of Cardiology, Copenhagen University Hospital - Rigshospitalet, Copenhagen, Denmark. ✉e-mail: michael.asger.andersen@regionh.dk

lowering LDL-C by up to 60% in persons on statin therapy[5,6]. Two large randomized controlled trials, FOURIER (evolocumab) and ODYSSEY OUTCOMES (alirocumab), have shown positive outcomes in terms of major adverse cardiovascular events (MACE) in persons receiving these anti-PCSK9 drugs[5,6]. However, the two drugs are not biosimilars but rather analog drugs and the two studies included different populations - stable atherosclerotic disease in the FOURIER study and acute coronary syndrome in the ODYSSEY study[5,6]. The two anti-PCSK9 drugs were developed and tested independently of each other, and therefore have not been directly compared in a clinical trial. Both drugs target *PCSK9* and were found to effectively lower LDL-C levels by approximately 60% in separate studies. Both drugs also demonstrated a reduction in non-fatal events, but neither showed an impact on fatal cardiovascular events[5,6].

Statin intolerance, frequently characterized by adverse effects such as muscle pain and weakness, and in rare cases, more serious side effects, is a significant factor driving patients toward alternative lipid-lowering therapies like *PCSK9* inhibitors. The prevalence of statin intolerance, influenced by both genetic and other factors, varies considerably among different populations, underscoring the need for diverse therapeutic options.

In February 2021, the Danish Medicines Council (DMC) concluded that the two anti-PCSK9 drugs were clinically equivalent and made a recommendation to preferentially use evolcumab based on price[7]. The decision to switch from one drug to the other applied to both new persons and persons already receiving alirocumab, with the goal of treating 80% of all persons with evolocumab. This switch presents a unique opportunity to compare the two drugs, which have never been directly compared in a clinical trial before.

The aim of this study is to investigate the effects of anti-PCSK9 treatment on the LDL-C levels in persons with hyperlipidemia, and to assess the impact of a mandated switch from alirocumab to evolocumab on these persons. First, we will assess the change in LDL-C following treatment initiation as well as after the mandated switch. Second, we will evaluate treatment retention, overall survival, and MACE among all anti-PCSK9 users. Lastly, we will explore factors predicting treatment switch, such as sex, age, comorbidities, and other drug allergies.

In this study, we find that anti-PCSK9 treatment significantly reduces LDL-C levels in individuals with hyperlipidemia. The mandated switch from alirocumab to evolocumab does not result in significant changes in LDL-C levels or adverse clinical outcomes, including major cardiovascular events and mortality. Both drugs demonstrate similar effectiveness and safety, suggesting they can be used interchangeably in managing hyperlipidemia. These findings provide valuable insights into the clinical equivalence of alirocumab and evolocumab, supporting their use in lipid management strategies.

## Methods
### Study design
This is a register-based cohort study of persons with dyslipidemia who received treatment with an anti-PCSK9 drug.

### Study cohort
The study included all persons who received anti-PCSK9 treatment in the Capital Region of Denmark from January 1, 2016, until July 26, 2022. Alirocumab and evolocumab were the only two anti-PCSK9 treatments used during the study period. Anti-PCSK9 treatment is provided to persons at no cost by hospitals in Denmark. The use of anti-PCSK9 treatment is limited by the national healthcare reimbursement policies and persons receiving anti-PCSK9 treatment must meet specific criteria. The Danish Medicines Council has divided the requirements for reimbursement for hypercholesterolemia treatment into two categories, namely familial hypercholesterolemia, and non-familial hypercholesterolemia. To be eligible for treatment, persons with hypercholesterolemia are required to meet the predefined criteria outlined by the council[7].

### Study variables
In this study, we obtained variables such as sex, age, vital status, laboratory values, prescriptions, administrations of medicines, allergies, and diagnosis and procedural codes describing the type and degree of the ASCVD from the Sundhedsplatformen program (EPIC EHR), see supplementary methods for full description. Familial hypercholesterolemia was defined using ICD-10 codes or LDL-C measurement greater than 5 mmol/L, see supplementary methods for ICD10 and ATC specifications.

### Outcome definitions
We considered persons who had stopped any anti-PCSK9 treatment for 30 days or more as having discontinued treatment; this included those who switched to another anti-PCSK9 drug after such a pause. MACE was defined as a myocardial infarction, a stroke, or death. Death was determined using the vital status from the SAP Web Intelligence (WebI); however, this source did not allow for differentiation between cardiovascular and non-cardiovascular causes of death.

### Statistical analysis
A generalized additive model (GAM) was used to examine the effect of a treatment initiation and switch in treatment on LDL-C levels, with each person treated as a random intercept. We fitted two models, one using all LDL-C measurements to evaluate the effect of anti-PCSK9 treatment and the effect of dosage. In another model, we only included LDL-C measurements within the range of 180 days prior to the mandatory switch and from 40 days post-switch up to 180 days, excluding measurements taken during treatment pauses. Covariates including age, sex, dose of anti-PCSK9, and other lipid-lowering medications were included in both models. To account for individual-level random effects, the "re" option of the GAM function was utilized. Penalized spline terms were also added for age to the model. For further details, please refer to supplementary information.

In a sensitivity analysis, we employed the t-test to assess the impact of transitioning from evolocumab to alirocumab and vice versa, both before and after the mandated switch date. For each group, we identified the most recent (closest to switch date) LDL-C measurement taken within a maximum of 180 days before the switch during ongoing treatment with an anti-PCSK9, as well as the LDL-C measurement taken between 40 and 180 days after the baseline and ongoing treatment with an anti-PCSK9.

In the second part of the study, we assessed treatment retention, ASCVD risk, and survival by setting up a cohort study and following individuals from the start of anti-PCSK9 treatment until death or the end of follow-up, whichever occurred first. Using a Poisson model, we modeled the number of first MACE and deaths, with person-days as an offset. For discontinuation, we included a random effect for each person and allowed for multiple events per person. All models were adjusted for variables such as the type of switch (first treatment initiation, switch before mandated date, switch after mandated date), calendar date, days on anti-PCSK9 treatment, sex, age, and presence of any comorbidity (Supplementary information). Continuous variables were included as penalized splines.

In the final part of the study, we examined all persons taking alirocumab on February 1, 2021, who were alive on December 31 of the same year. The first-year post-switch was chosen for analysis as it captured most switches, offering a focused period for evaluation. We used chi-squared and Wilcoxon rank sum tests to compare the group of persons who switched to evolocumab with the group who did not switch. Continuous variables are presented as means with standard deviations, and categorical data are presented as frequencies and percentages, unless otherwise stated. A two-tailed P value less than 0.05 was considered statistically significant, and 95% confidence intervals were provided. The study was conducted and reported in accordance with the Strengthening the Reporting of Observational Studies in Epidemiology (STROBE) recommendations[8]. Data management and statistical analysis were conducted using R version 4.0[9].

This study was conducted in accordance with all relevant ethical regulations concerning the use of human participants and data. Ethical approval for the study was obtained from the Regional Ethics Committee of the Capital Region of Denmark (approval number R-22034085). Given that this was a register-based cohort study utilizing data from electronic health records, informed consent was waived by the Ethics Committee. The waiver was granted because the study involved the analysis of existing, anonymized data, with no direct interaction with participants or interventions performed, minimizing potential risks. The study complied with the principles outlined in the Declaration of Helsinki.

### Reporting summary

Further information on research design is available in the Nature Portfolio Reporting Summary linked to this article.

## Results
### Participants

In this study, a total of 907 persons who had received treatment with an anti-PCSK9 agent at any point were included (see Fig. 1). The study population consisted of 457 men and 450 women, with a mean age of 61.3 years (range, 11–83 years). Of the 907 persons, 518 (57%) had familial hypercholesterolemia, including 302 (58%) in secondary prevention. At the time of the first anti-PCSK9 treatment, 133 (15%) did not receive any other lipid-modifying treatments, 389 (43%) received one drug, and 385 (42%) received two or more drugs. Statins were taken by 447 (49%) persons, with 221 (49%) on a high dose. Ezetimibe was taken by 609 (67%) persons, and fibrates were being taken by 87 (10%). Diabetes mellitus was present in 160 (18%) persons (see Table 1).

### LDL-C measurements

A total of 18,027 LDL-C measurements were included in the analysis, with a mean number of measurements per person of 20 (median 22, range 1–88).

The mean LDL-C level at the time of the first anti-PCSK9 treatment in familial hypercholesterolemia and non-familial hypercholesterolemia persons were 4.41 mmol/L (standard deviation [SD] 1.77 mmol/L) and 3.31 mmol/L (SD 1.50 mmol/L), respectively. After adjusting for sex, age, and concurrent lipid-lowering therapies, standard dose anti-PCSK9 treatment was associated with 49% (95% CI 48–50%) reduction in LDL-C levels. Compared with the standard dose of 140–150 mg, a dose of 75 mg was associated with a 12% higher LDL-C (95% CI, 8–16%). For the mandated switch analysis, LDL-C levels measured before the switch were 4% higher than those measured after the switch, (95% CI −3–11%). The sensitivity analysis revealed similar results, the mandated switch was associated with a mean change of 4% in LDL-C levels (95% CI −3–11%) among persons with measurements taken within the specified window. Further details are provided in Fig. 2.

### Outcomes

We followed 907 persons for a total of 1925 person-years and observed 288 discontinuations. No significant difference was found in the likelihood of discontinuation between different types of *PCSK9* inhibitors, with a risk ratio of 0.92 (95% CI 0.69–1.24) for evolocumab versus alirocumab. The risk of discontinuations was highest during the first 100 days following treatment initiation, as illustrated in Fig. 3. We observed 58 MACEs during 1844 person-years of follow-up and found no association between the type of *PCSK9* inhibitor and the risk of MACE, with a risk ratio of 1.06 (95% CI 0.60–1.88) both before and after the date of the mandated switch (Fig. 4). Similarly, no significant association was observed between the type of anti-PCSK9 treatment and the risk of death, with a risk ratio of 1.24 (95% CI 0.37–4.18). Of the 18 deaths recorded during the study, 12 occurred under evolocumab treatment and 6 under alirocumab, with 10 evolocumab-related deaths occurring outside of a hospital setting. Fewer than five persons had a diagnosis of a cardiovascular event according to ICD-10 coding, and the incidence of these events was comparable between the two treatment groups.

**Fig. 1 | Number of persons treated with Alirocumab and Evolocumab.** This figure displays the number of persons treated with alirocumab and evolocumab at a given date, as well as the total number of persons in treatment, from 2016 to 2022. PCSK9: proprotein convertase subtilisin/kexin type 9.

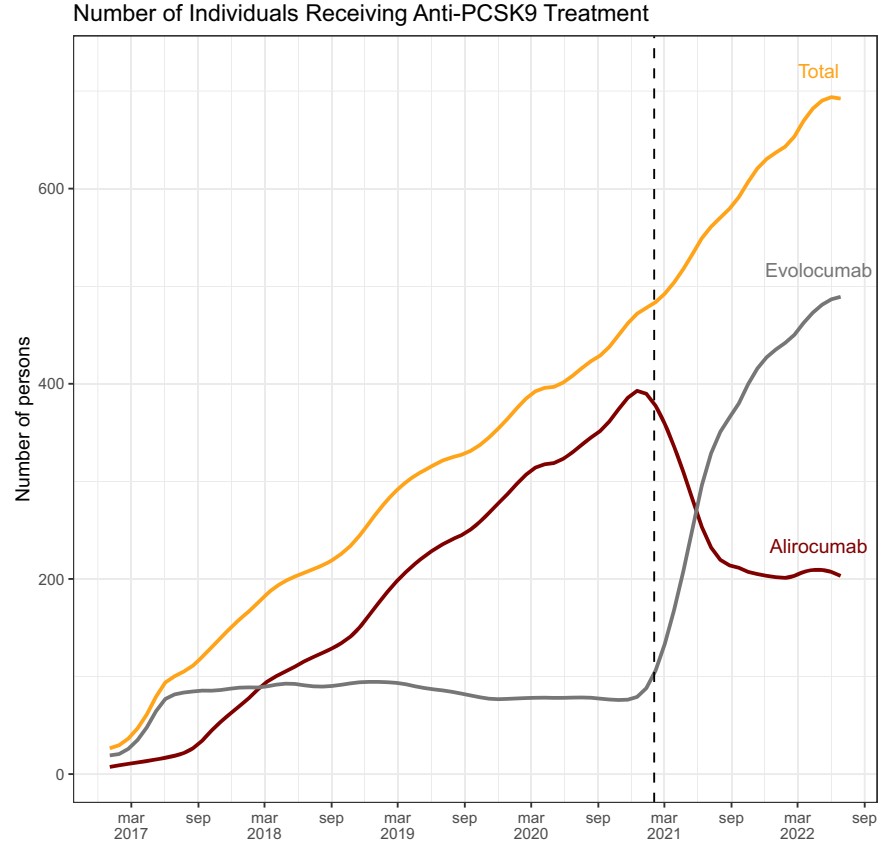

**Table 1 | Demographics and Baseline Characteristics**

| Characteristic | Familial hypercholesterolemia, *N* = 518[a] | Non-Familial hypercholesterolemia, *N* = 389[a] |
|---|---|---|
| Sex | | |
| Man | 223 (43%) | 234 (60%) |
| Woman | 295 (57%) | 155 (40%) |
| Age (years) | 60 (53, 68) | 65 (56, 71) |
| **Laboratory values** | | |
| Total cholesterol, mmol/L | 6.59 (5.50, 7.80) | 5.66 (4.92, 6.38) |
| Unknown | 63 | 117 |
| LDL levels, mmol/L | 4.36 (3.43, 5.41) | 3.41 (2.79, 4.00) |
| Unknown | 73 | 132 |
| HDL levels, mmol/L | 1.28 (1.08, 1.56) | 1.22 (0.99, 1.47) |
| Unknown | 63 | 116 |
| Triglycerides, mmol/L | 1.82 (1.26, 2.63) | 2.00 (1.25, 2.96) |
| Unknown | 64 | 117 |
| **Lipid modifying drugs** | | |
| Any statins | 272 (53%) | 175 (45%) |
| High dose statins | 136 (26%) | 85 (22%) |
| Moderate dose statins | 73 (14%) | 51 (13%) |
| Low dose statins | 63 (12%) | 39 (10%) |
| Fibrates | 44 (8.5%) | 43 (11%) |
| Bile acid sequestrants | 34 (6.6%) | 16 (4.1%) |
| Ezetimibe | 369 (71%) | 240 (62%) |
| Number of lipid modifying drugs | | |
| 0 drugs | 75 (14%) | 58 (15%) |
| 1 drug | 189 (36%) | 200 (51%) |
| 2+ drugs | 254 (49%) | 131 (34%) |
| **Comorbidities** | | |
| Atherosclerotic cardiovascular disease | 295 (57%) | 309 (79%) |
| Coronary heart disease | 252 (49%) | 274 (70%) |
| Atherosclerotic cerebrovascular disease | 56 (11%) | 47 (12%) |
| Peripheral artery disease | 52 (10%) | 61 (16%) |
| Chronic obstructive pulmonary disease | 48 (9.3%) | 53 (14%) |
| Heart failure | 45 (8.7%) | 58 (15%) |
| Rheumatoid arthritis | 15 (2.9%) | 14 (3.6%) |
| Cancer, ex. non-melanoma skin cancer | 46 (8.9%) | 28 (7.2%) |
| Chronic kidney disease | 17 (3.3%) | 12 (3.1%) |
| Hypertension | 265 (51%) | 277 (71%) |
| Diabetes mellitus | 75 (14%) | 85 (22%) |
| Diabetes mellitus with organ damage | 51 (9.8%) | 56 (14%) |

[a]*n* (%); Median (IQR).

This table presents the demographic characteristics of the study population at the time of the first anti-PCSK9 treatment, including lipid measurements, comorbidities, and lipid-lowering medications. Atherosclerotic cardiovascular disease in this context includes all ICD-10 codes corresponding to coronary heart disease, cerebrovascular disease, and peripheral artery disease. Statins were classified as follows: Simvastatin (low: 0 mg, moderate: 20 mg, high: 80 mg), Pravastatin (low: 0 mg, moderate: 40 mg), Fluvastatin (low: 0 mg, moderate: 80 mg), Atorvastatin (low: 0 mg, moderate: 10 mg, high: 40 mg), and Rosuvastatin (low: 0 mg, moderate: 5 mg, high: 20 mg). HDL-C: low-density lipoprotein cholesterol, LDL-C: low-density lipoprotein cholesterol.

Additional analyses, which included an interaction term between calendar period and type of anti-PCSK9 treatment and were compared using ANOVA testing, did not reveal any statistically significant differences, indicating no variation in outcomes based on treatment type over time.

**Characterization of switchers**

As of 1 February 2021, 380 persons were receiving alirocumab treatment. Of these, 3 persons died before 31 December 2021, leaving 377 individuals in this cohort. Among them, 195 (52%) switched to evolocumab, while the remaining 182 (48%) continued with alirocumab. Notably, 8 of the 182 non-switchers had previously receieved evolocumab.

Our study revealed that a higher proportion of women did not switch anti-PCSK9 treatments during the first year of the study compared to men. We found that 55% of women did not switch compared to 41% of men. This difference was statistically significant (*p*-value = 0.007). Interestingly, we also found that non-switchers were less likely to have received any statins and other lipid modifying treatments before initiation of anti-PCSK9 therapy compared to those who did switch treatments. These findings are detailed in Table 2.

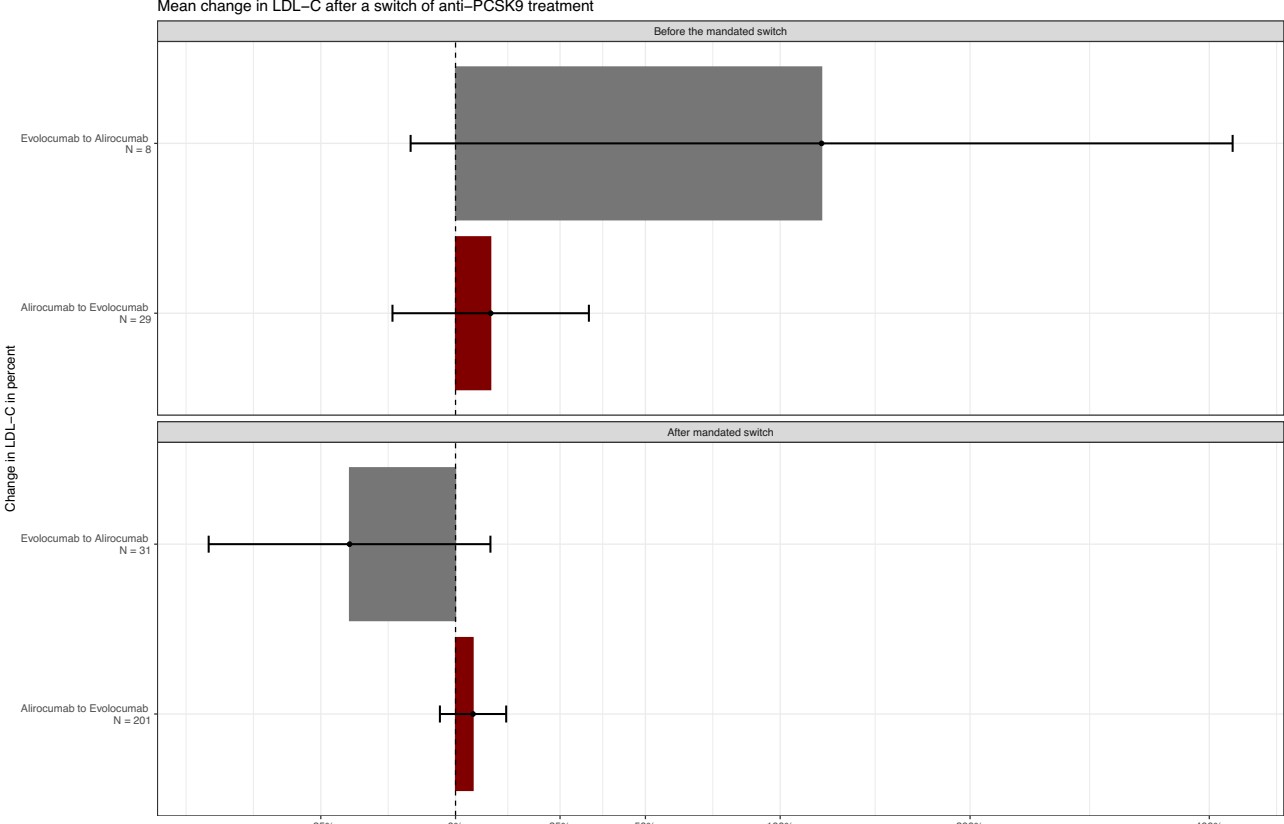

**Fig. 2 | Mean Change in Percent Following Switch from One Anti-PCSK9 Treatment to Another.** This figure displays the mean change in percent following a mandated switch from one anti-PCSK9 treatment to another, along with 95% confidence intervals represented by black error bars. The upper panel depicts persons who switched before the mandate, while the lower panel shows those who switched afterward. The y-axis shows the type of anti-PCSK9 treatment received before and after the switch, while the x-axis represents the mean change in percent. The red boxes represent the mean change in percent for persons who switched from alirocumab to evolocumab, and the gray boxes represent the mean change in percent for persons who switched from evolocumab to alirocumab. The number of persons included in each analysis is indicated on each line, before the mandated switch evolocumab to alirocumab $n = 8$, alirocumab to evolocumab $n = 29$, after the mandated switch evolocumab to alirocumab $n = 31$, alirocumab to evolocumab $n = 201$. PCSK9 proprotein convertase subtilisin/kexin type 9, HDL-C low-density lipoprotein cholesterol, LDL-C low-density lipoprotein cholesterol, N number.

**Fig. 3 | Relative risk of discontinuing Anti-PCSK9 treatment.** This graph displays the relative risk of discontinuing anti-PCSK9 treatment over time. The x-axis represents the number of days since treatment initiation, and the y-axis displays the relative risk of discontinuing treatment. The analysis includes a total of $n = 18,027$ measurements in 907 persons. The shaded area around the line indicates the range within which the true relative risk is expected to fall with 95% certainty. PCSK9: proprotein convertase subtilisin/kexin type 9.

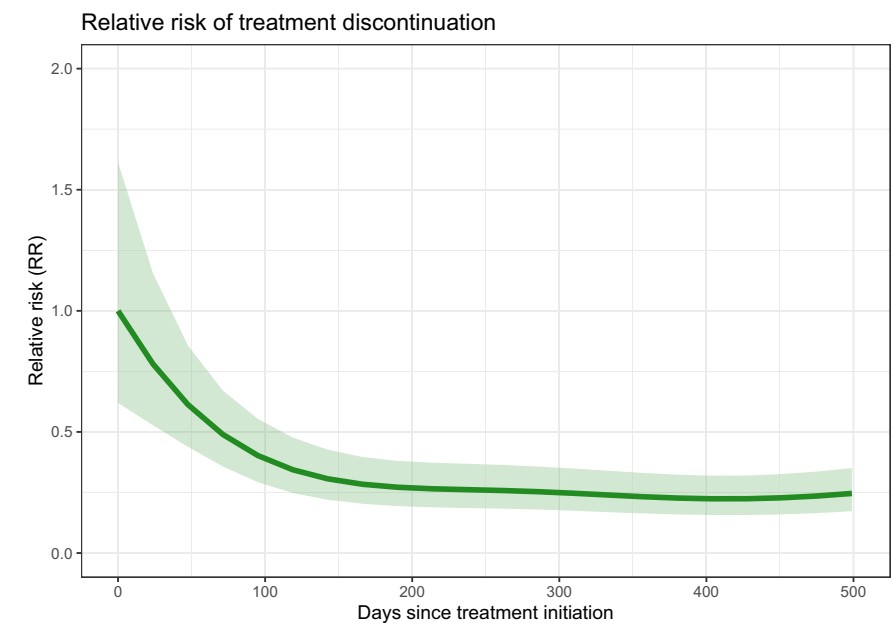

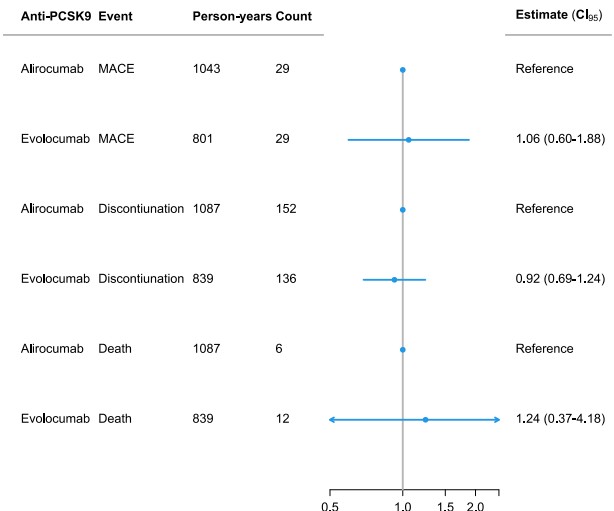

| Anti-PCSK9 | Event | Person-years | Count | Estimate (CI95) |
|---|---|---|---|---|
| Alirocumab | MACE | 1043 | 29 | Reference |
| Evolocumab | MACE | 801 | 29 | 1.06 (0.60-1.88) |
| Alirocumab | Discontinuation | 1087 | 152 | Reference |
| Evolocumab | Discontinuation | 839 | 136 | 0.92 (0.69-1.24) |
| Alirocumab | Death | 1087 | 6 | Reference |
| Evolocumab | Death | 839 | 12 | 1.24 (0.37-4.18) |

**Fig. 4 | Incidence rates for persons treated with Alirocumab or Evolocumab.** This table presents the number of person-years, events, and incidence rates for persons treated with either alirocumab or evolocumab, adjusted for sex, calendar period, and treatment duration.

## Discussion

In this observational study of 907 persons with dyslipidemia, we observed a significant decrease in LDL-C levels after starting treatment with anti-PCSK9. However, when a subset of these persons was required to switch from alirocumab to evolocumab, we did not observe any differences in LDL-C levels or clinical outcomes, including MACE and mortality.

The FOURIER trial showed that the *PCSK9* inhibitor evolocumab had a cardiovascular benefit compared to placebo[5]. However, a subsequent study used information from the Clinical Study Report to restore the mortality data and found that the cause of death adjudicated by the FOURIER clinical events committee differed from that declared by the local clinical investigator for 41.4% of deaths[5,10]. Our study also observed more deaths in the evolocumab group compared to the alirocumab group; however, the relative increase in mortality was still non-significant, and our study was not designed to establish such causality due to its small sample size.

Although this study is limited by its observational nature, it offers a unique opportunity to compare the effects of two different anti-PCSK9 treatments through a case-crossover design. The mandated switch from one treatment to another was not based on the discretion of the physician or person, allowing for a robust comparison of the treatments' effects. Our findings, which demonstrate no significant difference between the reduction of LDL-C and hard endpoints after the mandated switch, suggest that alirocumab and evolocumab may have similar effectiveness in reducing the risk of cardiovascular events. These findings align with previous reports that have indicated similar effects of the two treatments on LDL-C levels and cardiovascular outcomes[5,6,11].

Lastly, our study chose a 30-day cutoff for defining treatment discontinuation to account for the potential loss of treatment effect due to prolonged interruption in treatment. We explored different thresholds, such as 20-day and 50-day cutoffs, which resulted in slightly different numbers of discontinuations but the estimates were similar for the models (data not shown). The study results are based on data from the electronic health records, which are comprehensive for many variables and used for the national registries[12]. However, as with any observational study, it is important to keep in mind that there may be some inaccuracies in reporting. One important consideration when interpreting the results of our study is the potential for bias due to the role of the physician and patient's attitudes towards switching treatment. The observed data shows that only 50% of the persons switch their treatment. It is possible that physicians may have a bias towards maintaining the current treatment, or that persons may be hesitant to switch treatments due to concerns about side effects or unfamiliarity with

### Table 2 | Comparison of Persons Who Switched Treatments

| Characteristic | Non-switchers, N = 182[a] | Switchers, N = 195[a] | p-value[b] |
|---|---|---|---|
| Sex | | | 0.007 |
| Woman | 110 (60%) | 91 (47%) | |
| Man | 72 (40%) | 104 (53%) | |
| Age (years) | 63 (54, 69) | 60 (54, 67) | 0.077 |
| Familial hypercholesterolemia | | | >0.9 |
| Familial hypercholesterolemia | 109 (60%) | 118 (61%) | |
| Non-Familial hypercholesterolemia | 73 (40%) | 77 (39%) | |
| **Lipid modifying drugs** | | | |
| Any statins | 74 (41%) | 113 (58%) | <0.001 |
| High dose statins | 30 (16%) | 50 (26%) | 0.030 |
| Moderate dose statins | 29 (16%) | 29 (15%) | 0.8 |
| Low dose statins | 15 (8.2%) | 34 (17%) | 0.008 |
| Fibrates | 23 (13%) | 26 (13%) | 0.8 |
| Bile acid sequestrants | 7 (3.8%) | 9 (4.6%) | 0.7 |
| Ezetimibe | 118 (65%) | 137 (70%) | 0.3 |
| Number of lipid modifying drugs | | | 0.004 |
| 0 drugs | 39 (21%) | 19 (9.7%) | |
| 1 drug | 73 (40%) | 78 (40%) | |
| 2+ drugs | 70 (38%) | 98 (50%) | |
| **Comorbidities** | | | |
| Atherosclerotic cardiovascular disease | 115 (63%) | 132 (68%) | 0.4 |
| Coronary heart disease | 102 (56%) | 112 (57%) | 0.8 |
| Atherosclerotic cerebrovascular disease | 20 (11%) | 20 (10%) | 0.8 |
| Peripheral artery disease | 18 (9.9%) | 18 (9.2%) | 0.8 |
| Hypertension | 119 (65%) | 115 (59%) | 0.2 |
| Diabetes mellitus | 41 (23%) | 30 (15%) | 0.076 |
| Diabetes mellitus with organ damage | 22 (12%) | 22 (11%) | 0.8 |
| Heart failure | 18 (9.9%) | 16 (8.2%) | 0.6 |
| Chronic obstructive pulmonary disease | 23 (13%) | 17 (8.7%) | 0.2 |
| Rheumatoid arthritis | 9 (4.9%) | 5 (2.6%) | 0.2 |
| Cancer, ex. non-melanoma skin cancer | 12 (6.6%) | 15 (7.7%) | 0.7 |
| Chronic kidney disease | 5 (2.7%) | 3 (1.5%) | 0.5 |
| **Allergies** | | | |
| Number of allergies | | | 0.13 |
| 0 | 79 (43%) | 107 (55%) | |
| 1 | 40 (22%) | 35 (18%) | |
| 01-mar | 44 (24%) | 33 (17%) | |
| 4+ | 19 (10%) | 20 (10%) | |
| Statin allergy | 58 (32%) | 57 (29%) | 0.6 |
| Ezetimibe allergy | 16 (8.8%) | 18 (9.2%) | 0.9 |

[a]n (%); Median (IQR)
[b]Pearson's Chi-squared test; Wilcoxon rank sum test; Fisher's exact test
This table compares the demographic and clinical characteristics of persons who switched treatments during the first year of the study to those who did not switch. Variables such as age, gender, baseline lipid measurements, and comorbidities are included. Chi-squared and Wilcoxon rank sum tests were used to compare the group of persons who switched to evolocumab with the group who did not switch. Atherosclerotic cardiovascular disease in this context includes all ICD-10 codes corresponding to coronary heart disease, cerebrovascular disease, and peripheral artery disease. Statins were classified as follows: Simvastatin (low: 0 mg, moderate: 20 mg, high: 80 mg), Pravastatin (low: 0 mg, moderate: 40 mg), Fluvastatin (low: 0 mg, moderate: 80 mg), Atorvastatin (low: 0 mg, moderate: 10 mg, high: 40 mg), and Rosuvastatin (low: 0 mg, moderate: 5 mg, high: 20 mg).

new treatments. This bias could potentially impact the results of the study and should be considered when interpreting the findings. In addition, certain important variables such as smoking and body mass index were not available for analysis, which could potentially influence the results if there are significant differences between treatment groups. The first day of an anti-PSCK9 treatment is also notable. Upon review of person charts, it was discovered that a small number of persons received anti-PCSK9 treatment without proper documentation. This bias can have led to an underestimation of the treatment effect, as some persons LDL-C levels were lowered by the treatment but were not recorded as receiving it. Despite this, our study found a reduction of 49%, which is lower compared to previous clinical trials but similar to other real-world studies[5,6,11,13]. Nevertheless, it is unlikely that this bias significantly impacted the estimate for the switch.

Finally, we analyzed the data of all persons taking alirocumab on February 1, 2021, who were still alive on December 31 of that year. In a recent meta-analysis on statin intolerance, the authors identified several risk factors that were significantly associated with statin intolerance, including female sex. This finding is consistent with our observation where we identified a statistically significant disparity in gender distribution between those who switched medications and those who did not. This pattern of sex difference aligns with the broader understanding of medication tolerance and patient responses, emphasizing the need for personalized treatment approaches in lipid management[14]. Interestingly, our data did not reveal a difference in statin allergy between those who switched *PCSK9* inhibitors and those who did not, suggesting other factors may influence this decision. Notably, we observed marked differences in practices across hospitals, indicating varying adherence to guidelines issued by the Danish Medicines Council. This variation may partly explain the observed differences in medication-switching behavior.

In conclusion, our study investigated the effects of a nationwide switch from alirocumab to evolocumab on LDL-C levels, employing two distinct analytical methods. Our findings consistently revealed no significant change in LDL-C levels following this switch. While negative expectations or nocebo effects are often a concern in treatment switch studies, the objective and difficult-to-influence nature of the study outcomes, in this case, suggest that they were not a factor[12]. These findings indicate that there is no evidence of a change in LDL-C levels because of the nationwide mandatory switch from alirocumab to evolocumab. This leads us to conclude that alirocumab and evolocumab are comparably effective in lowering LDL-C levels, thus endorsing the feasibility of switching between these treatments without compromising efficacy.

## Data availability
Due to restrictions imposed by the Danish Data Protection Agency and the nature of the data being derived from national health registries, the source data underlying the graphs and charts presented in this manuscript cannot be publicly deposited or shared. Access to the raw data and source data is governed by strict confidentiality agreements and is not permitted outside of approved research projects. Researchers interested in accessing the data may submit a formal request to the Capital Region of Denmark, subject to approval and under the conditions set by the relevant ethical and legal bodies.

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

## Acknowledgements
We would like to express our sincere gratitude to all the persons in the Capital Region who participated in this study. This study was funded by the Department of Clinical Pharmacology at Bispebjerg and Frederiksberg Hospital.

## Author contributions
T.S.P. conceptualized the study idea. T.S.P. and M.A.A. designed the study and the methods. M.A.A., A.H.A., L.E.B., E.J.S., and T.S.P. interpreted the results. M.A.A. prepared the original draft of the manuscript. All authors contributed to writing and approving the final version of the manuscript.

## Competing interests
The authors declare no competing interests.
