## [Peer Review File · Communications Medicine]

Reviewers' comments:

Reviewer #1 (Remarks to the Author):

The submitted manuscript addresses the safety and efficacy of switching between PCSK9 inhibitors in 907 patients in Denmark. The study was conducted as a registry-based cohort of individuals with dyslipidemia receiving PCSK9 inhibitor therapy during a time in which a recommendation was made in Denmark for which PCSK9 inhibitor to be on based on unit price. The primary aims of the study were to assess the change in LDL-C before and after the switch, to evaluate adherence, survival and MACE and to identify factors associated with treatment switching. The authors concluded that standard dose treatment was associated with a 49% lowering of LDL-C. Switching PCSK9 inhibitor therapy was associated with a 4% lowering of the LDL-C which was nonsignificant. There was no significant difference in adherence between groups. There was no association between MACE and death and type of PCSK9 inhibitor. The authors also evaluated characteristics associated with switching, noting that women were less likely to switch in the first year and those who did not switch were less likely to be received other cholesterol lowering therapies ahead of PCSK9 inhibitor initiation.

Overall, the authors use a national recommendation to assess the impact of switching PCSK9 inhibitors in a patient population with dyslipidemia which has not been previously studied in a retrospective cohort study design. While the study is interesting and novel, there are a number of areas for improvement.

1. In line 78, the authors note that their second objective is to assess treatment retention, overall survival and MACE. In both this section and the results section, it is unclear what population this is being assessed in. Is this among all PCSK9 users? After the switch?
2. Line 115 mentions no association between type of PCSK9 inhibitor and risk of MACE, again it is unclear what population this is studied in – individuals that switched?
3. The third objective of the study (line 79) is to assess factors predicting treatment switch. The relevance of this objection is not clear in the setting of a what is described as a nationally mandated goal of having 80% of all patients on Evolocumab.
4. There is confusing language in the conclusions regarding the treatment switch. It is both noted as a mandated switch, not at the discretion of the physician or patient but also that physicians and patients could opt not to switch. This adds to the confusion of the 3rd objective of this study.
5. In Table 2, is the study population patients that did not switch? How are the person-years attributed if the patient was switched to a different PCSK9 inhibitor.
6. Table 3 evaluates the characteristics of patients who switched vs did not switch in the first year of the study. It's not clear why this time period was selected when patients included over a longer duration.
7. Figure 2 need additional description. The number of patients analyzed before and after the switch are different per group. It's not clear why that is the case. Additionally, the display of this figure on in the main document has the title cut off. There is also overlap of the Y axis label and the treatment groups.

Reviewer #2 (Remarks to the Author):

Dear Editors,

The authors attempt to present a direct head-to-head comparison between the two PCSK9 inhibitors (PCSK9i) that are currently on the market. It's a topic of interest to many cardiologists and as the authors note, the Danish Medical Council's mandate to switch from alirocumab to evolcumab offers a rich opportunity for exactly this kind of evaluation.

There are meaningful conclusions to be made from this paper, primarily that there is no difference between LDL and clinical outcomes between the two PCSK9i. However, the overall organization of the results appears to be chaotic and confusing.

It is more intuitive to discuss characteristics of both patient populations first - Table 1 for those on PCSK9i (FH vs non-FH) followed by Table 3 for switchers vs non-switchers. Unclear why authors compared FH vs non-FH patients at all in Table 1; it could as well have been a summary of the 907 patients without any comparisons since no meaningful conclusion is made with the comparison.

After patient characteristics of both patient groups, I would prefer to hear more about trends in time - risk of discontinuation - followed by results on LDL change and finally the results on clinical outcomes at the end (i.e. Table 2, which is actually a figure).

It is of interest that only 49% of the study population was on a statin, and as such it may be worth expounding on the side effects of statins at the end of paragraph 2 in Background; statin intolerance is a big reason why patients go on to PCSK9i. I would also be interested in hearing more discussion about why people not on statins are also less likely to switch PCSK9i (especially as there is no difference in statin allergy between the switchers and non-switchers). Is it somehow related to how the DMC mandate is implemented; i.e. how much of a choice do patients and providers get in switching? Does *not* switching require a special approval process and take into account prior statin intolerance?

Please differentiate between all-cause mortality vs cardiovascular mortality, or state that it could not be differentiated if the death data does not provide those details.

Based on supplementary data it appears that Atherosclerotic CV disease (295 in FH patients, 309 in non-FH patients) encompasses all the ICD-10 codes for coronary heart disease, cerebrovascular disease and peripheral artery disease. This needs to be made explicit in tables 1 & 3 (either by indentation of the subcategories or in the table footnote). I would also recommend keeping the text in line with the data you have (i.e. 57% of FH patients were on PCSK9i for secondary prevention, rather than the extrapolated 43% were on it for primary prevention).

Unclear what is the added value of Figure 1 in supplements over Figure 2 in the manuscript itself.

Grammar/spelling: authors tend to switch between British English and American English spelling (for example, dyslipidemia vs dyslipidaemia). Please keep it consistent.

Other minor grammar and sentence structure errors/suggestions:

Abstract: "The mandated switch..." to "A mandated switch..."

Background, Paragraph 3:

“Although the two drugs are not biosimilars but analogue drugs and the studies included different populations, respectively stable atherosclerotic persons in the FOURIER study and persons with acute coronary syndrome in the ODYSSEY study. Consequently, the two anti-PCSK9...”

To

“However, the two drugs are not biosimilars but rather analogue drugs and the two studies included different populations - stable atherosclerotic disease in the FOURIER study and acute coronary syndrome in the ODYSSEY study. The two anti-PCSK9...”

Background, paragraph 4: “... made a recommendation on which anti-PCSK9 to be used based on price.” To “... made a recommendation to preferentially use evolcumab based on price.”

Discussion, paragraph 4, last line: “(data now shown)” to “(data not shown)”

Discussion, paragraph 5:

“...smoking and body mass index were not available for analysis, which could potentially influence the results if they were found to be related to treatment switching.”

To

“...smoking and body mass index were not available for analysis, which could potentially influence the results if there are significant differences between treatment groups.”

Methods, Study variables: ASCD to ASCVD.

Methods, Study variables:

“Familial hypercholesterolemia was defined using ICD-10 codes (Appendix). Additionally, we defined all persons suspected of familial hypercholesterolemia (with an ICD-10 code) and any LDL-C measurement greater than 5 mmol/L as having familial hypercholesterolemia.”

To

“Familial hypercholesterolemia was defined using ICD-10 codes (Appendix) or any person with LDL-C measurement greater than 5 mmol/L.”

Title page

Title: Effectiveness and Safety of Anti-PCSK9 Treatment in Persons with Hyperlipidemia: A Register-Based Cohort Study

Authors: Michael Asger Andersen¹, Anne Helms Andreasen², Lia Evi Bang³, Espen Jimenez Solem¹, and Tonny Studsgaard Petersen¹

1 Department of Clinical Pharmacology, Copenhagen University Hospital - Bispebjerg and Frederiksberg Hospital, Copenhagen, Denmark

2 Center for Clinical Research and Prevention, Copenhagen University Hospital - Bispebjerg and Frederiksberg, Copenhagen, Denmark

3 Department of Cardiology, Copenhagen University Hospital - Rigshospitalet, Copenhagen, Denmark

Corresponding Author: Michael Asger Andersen

Address: Bispebjerg Bakke 23, 2400 København NV

Email: michael.asger.andersen@regionh.dk

Phone number: +45 38635102

Abstract

Addressing dyslipidaemia, a condition involving abnormal amounts of lipids in the blood, is crucial due to its strong association with cardiovascular disease. Anti-protein convertase subtilisin/kexin type 9 (PCSK9) drugs like alirocumab and evolocumab offer promising therapeutic solutions. Here we show, from our register-based cohort study in the Capital Region of Denmark (2016-2022), the efficacy and safety of these drugs in 907 persons with dyslipidaemia. Upon treatment initiation, PCSK9 inhibitors showed a substantial 49% reduction in low-density lipoprotein cholesterol (LDL-C) levels. The mandated switch from alirocumab to evolocumab resulted in no significant difference in LDL-C levels or adverse clinical outcomes, such as major cardiovascular events or mortality. Thus, we conclude both alirocumab and evolocumab can be safely and effectively used interchangeably for treating dyslipidaemia.

Keywords: dyslipidaemia, atherosclerotic cardiovascular disease (ASCVD), statins, proprotein convertase subtilisin/kexin type 9 (PCSK9), alirocumab, evolocumab

Background

Low-density lipoprotein cholesterol (LDL-C) has been consistently linked to the development of atherosclerotic cardiovascular disease (ASCVD) ¹. Reducing LDL-C levels can inhibit the progression of atherosclerotic plaques and lower the risk of ASCVD ². It has been estimated that the risk of ASCVD can be reduced by as much as 22% over 5 years with each 1 mmol/L reduction in LDL-C ².

Statins are widely accepted and effective treatment for dyslipidemia and have proven to effectively lower LDL-C levels in numerous randomized clinical trials ³. However, some persons may require additional lipid-lowering medications even when taking high-dose statins ^{2,3}.

Proprotein convertase subtilisin/kexin type 9 (PCSK9) plays a role in regulating LDL-C levels by binding to LDL receptors and inhibiting their recycling ⁴. Inhibition of PCSK9 can increase the recycling of LDL receptors, leading to a decrease in LDL-C in plasma ⁴.

Alirocumab and evolocumab are fully humanized monoclonal antibodies that bind free plasma PCSK9, promoting degradation of PCSK9. PCSK9 antibodies (anti-PCSK9) are capable of lowering LDL-C by up to 60% in persons on statin therapy ^{5,6}. Two large randomized controlled trials, FOURIER (evolocumab) and ODYSSEY OUTCOMES (alirocumab), have shown positive outcomes in terms of major cardiovascular events (MACE) in persons receiving these anti-PCSK9 drugs ^{5,6}. However, the two drugs are not biosimilars but rather analogue drugs and the two studies included different populations - stable atherosclerotic disease in the FOURIER study and acute coronary syndrome in the ODYSSEY study ^{5,6}. The two anti-PCSK9 drugs were developed and tested independently of each other, and therefore have not been directly compared in a clinical trial. Both drugs target PCSK9 and were found to effectively lower LDL-C levels by approximately 60% in separate

studies. Both drugs also demonstrated a reduction in non-fatal events, but neither showed an impact on fatal cardiovascular events ^{5,6}.

In February 2021, the Danish Medicines Council (DMC) concluded that the two anti-PCSK9 drugs were clinically equivalent and made a recommendation to preferentially use evolocumab based on price ⁷. The decision to switch from one drug to the other applied to both new persons and persons already receiving alirocumab, with the goal of treating 80% of all persons with evolocumab. This switch presents a unique opportunity to compare the two drugs, which have never been directly compared in a clinical trial before.

The aim of this study is to investigate the effects of anti-PCSK9 treatment on the LDL-C levels in persons with hyperlipidemia, and to assess the impact of a mandated switch from alirocumab to evolocumab on these persons. Firstly, we will assess the change in LDL-C following treatment initiation as well as after the mandated switch. Secondly, we will evaluate treatment retention, overall survival and MACE. Lastly, we will explore factors predicting treatment switch, such as sex, age, comorbidities and other drug allergies.

Overall, this study aims to gain a better understanding of the effects of anti-PCSK9 treatment on LDL-C levels and the potential impact of a mandated switch from alirocumab to evolocumab in persons with hyperlipidemia.

Results

Participants

In this study, a total of 907 persons were included who had received treatment with an anti-PCSK9 at any point. Figure 1. The study population consisted of 457 men and 450 women, with a mean age of 61.3 years (range, 11–83 years). Of the 907 persons, 518 (57%) had familial hypercholesterolemia, and of those 216 (43%) were on it for primary prevention. At the time of first anti-PCSK9 treatment, 133 (15%) did not receive any other lipid-modifying

treatments, 389 (43%) received one drug, and 385 (42%) received two or more drugs. Statins were being taken by 447 (49%) persons, with 221 (49%) on a high dose. Ezetimibe was being taken by 609 (67%) persons, and 87 (10%) were taking fibrates. Diabetes mellitus was present in 160 (18%) persons, Table 1.

LDL-C measurements

A total of 18,027 LDL-C measurements were included in the analysis, with a mean measurements per person of 20 (median 22, range 1–88). The mean LDL-C level at time of first anti-PCSK9 treatment in familial hypercholesterolemia and non- familial hypercholesterolemia persons were 4.41 mmol/L (standard deviation [SD] 1.77 mmol/L) and 3.31 (SD 1.50 mmol/L), respectively. After adjusting for sex, age, and concurrent lipid-lowering therapies, standard dose anti-PCSK9 treatment was associated with 49% (95% CI 48%–50%) reduction in LDL-C levels. Compared with the standard dose of 140-150 mg, a dose of 75 mg was associated with a 12% higher LDL-C (95% CI, 8%–16). For the mandated switch analysis, LDL-C levels measured before the switch were 4% higher than those measured after the switch, (95% CI -3%–11%). The sensitivity analysis revealed similar results, the mandated switch was associated with a mean change of 4% (95% CI -3%–11%) in LDL-C levels for persons with measurements taken within the specified window. Further details are provided in Figure 2 and the Appendix.

Outcomes

We followed 907 persons for a total of 1925 person-years and observed 288 discontinuations. Table 2. We did not find any significant difference in the likelihood of discontinuation between different types of PCSK9 inhibitors, with a risk ratio of 0.92 (95% CI 0.69–1.24) for evolocumab versus alirocumab. The risk of discontinuations was highest within the first 100 days of treatment, as seen in Figure 3. We also observed 58 MACEs during 1844 person-

years of follow-up but found no association between the type of PCSK9 inhibitor and the risk of MACE, with a risk ratio of 1.06 (95% CI 0.60–1.88). Similarly, we did not find any significant association between the type of anti-PCSK9 treatment and the risk of death, with a risk ratio of 1.24 (95% CI 0.37-4.18). During the study period, 18 persons passed away, 12 of which were under evolocumab treatment and 6 under alirocumab treatment. Ten persons receiving evolocumab treatment died outside of hospital setting. There were fewer than 5 persons who had a diagnosis of cardiovascular event in the ICD-10, and the incidence of cardiovascular events was similar between the two treatment groups.

We conducted additional analyses to determine whether the relationship between treatment type and outcomes varied over time by adding an interaction term between calendar period and type of anti-PCSK9 and comparing the models using ANOVA testing. However, the results of these analyses did not reveal any statistically significant differences, indicating that there were no differences in outcomes based on treatment type over time. For the persons with familial hypercholesterolemia in primary prevention, we found a total of 216 persons who were followed for 410 persons years during which we observed two MACEs.

Characterization of switchers

On 1 February 2021, 380 persons were treated with alirocumab of whom 3 persons died before December 31, 2021. Out of these 377 persons, 195 (52%) switched to evolocumab while the remaining 182 (48%) did not switch. Out of the 182 persons who did not switch, 8 had previously been treated with evolocumab.

Our study revealed that a higher proportion of women did not switch anti-PCSK9 treatments during the first year of the study compared to men. We found that 55% of women did not switch compared to 41% of men. This difference was statistically significant (p-value = 0.007). Interestingly, we also found that non-switchers were less likely to have received any

statins and other lipid modifying treatments before initiation of anti-PCSK9 therapy compared to those who did switch treatments. These results are presented in Table 3.

Discussion

In this observational study of 907 persons with dyslipidemia, we observed a significant decrease in LDL-C levels after starting treatment with anti-PCSK9. However, when a subset of these persons was required to switch from alirocumab to evolocumab, we did not observe any differences in LDL-C levels or clinical outcomes, including MACE and mortality.

The FOURIER trial showed that the PCSK9 inhibitor evolocumab had a cardiovascular benefit compared to placebo ⁵. However, a subsequent study used information from the Clinical Study Report to restore the mortality data and found that the cause of death adjudicated by the FOURIER clinical events committee differed from that declared by the local clinical investigator for 41.4% of deaths ^{5,10}. Our study also observed more deaths in the evolocumab group compared to the alirocumab group, however the relative increase in mortality was still non-significant, and our study was not designed to establish such causality due to its small sample size.

Although this study is limited by its observational nature, it offers a unique opportunity to compare the effects of two different anti-PCSK9 treatments through a case-crossover design. The mandated switch from one treatment to another was not based on the discretion of the physician or person, allowing for a robust comparison of the treatments' effects. Our findings, which demonstrate no significant difference between the reduction of LDL-C and hard endpoints after the mandated switch, suggest that alirocumab and evolocumab may have similar effectiveness in reducing the risk of cardiovascular events. These findings align with previous reports that have indicated similar effects of the two treatments on LDL-C levels and cardiovascular outcomes ^{5,6,11}.

Lastly, our study chose a 30-day cutoff for defining treatment discontinuation to account for the potential loss of treatment effect due to prolonged interruption in treatment. We explored different thresholds, such as 20-day and 50-day cutoffs, which resulted in slightly different numbers of discontinuations but the estimates were similar for the models (data not shown). The study results are based on data from the electronic health records, which are comprehensive for many variables and used for the national registries¹². However, as with any observational study, it is important to keep in mind that there may be some inaccuracies in reporting. One important consideration when interpreting the results of our study is the potential for bias due to the role of the physician and patient's attitudes towards switching treatment. The observed data shows that only 50% of the persons switch their treatment. It is possible that physicians may have a bias towards maintaining the current treatment, or that persons may be hesitant to switch treatments due to concerns about side effects or unfamiliarity with new treatments. This bias could potentially impact the results of the study and should be considered when interpreting the findings. In addition, certain important variables such as smoking and body mass index were not available for analysis, which could potentially influence the results if there were significant differences between treatment groups. The first day of an anti-PCSK9 treatment is also notable. Upon review of person charts, it was discovered that a small number of persons received anti-PCSK9 treatment without proper documentation. This bias can have led to an underestimation of the treatment effect, as some persons LDL-C levels were lowered by the treatment but were not recorded as receiving it. Despite this, our study found a reduction of 49%, which is lower compared to previous clinical trials but similar to other real-world studies^{5,6,11,13}. Nevertheless, it is unlikely that this bias significantly impacted the estimate for the switch.

Finally, we analyzed the data of all persons taking alirocumab on February 1, 2021, who were still alive on December 31 of that year. In a recent meta-analysis on statin intolerance, the

authors identified several risk factors that were significantly associated with statin intolerance, including female sex. This finding is consistent with our observation of a statistically significant sex difference between switchers and non-switchers ¹⁴.

In conclusion, the current study on the impact of a non-medical switch on LDL-C levels used two different approaches and found no change in LDL-C levels in either case. While negative expectations or nocebo effects are often a concern in treatment studies, the objective and difficult-to-influence nature of the study outcomes in this case suggest that they were not a factor ¹². These findings indicate that there is no evidence of a change in LDL-C levels because of the nationwide mandatory switch from alirocumab to evolocumab. Based on the results of this study, both alirocumab and evolocumab appear to be equally effective at lowering LDL-C levels and it is possible to switch between the two treatments without loss of efficacy.

Methods

Study design

This is register-based cohort study of persons with dyslipidemia who received treatment with an anti-PCSK9 drug.

Study cohort

The study included all persons who received anti-PCSK9 treatment in the Capital Region of Denmark from January 1, 2016, until July 26, 2022. Alirocumab and evolocumab were the only two anti-PCSK9 treatment used in the study period. Anti-PCSK9 treatment is provided to persons at no cost by hospitals in Denmark. The use of anti-PCSK9 treatment is limited by the national healthcare reimbursement policies and persons who receive anti-PCSK9 treatment are required to meet specific treatment requirements. The Danish Medicines Council has divided the requirements for reimbursement for hypercholesterolemia treatment

into two categories, namely familial hypercholesterolemia, and non-familial hypercholesterolemia. To be eligible for treatment, persons with hypercholesterolemia are required to meet the predefined criteria outlined by the council ⁷.

Study variables

In this study, we obtained data variables such as sex, age, vital status, laboratory values, prescriptions, administrations of medicines, allergies, and diagnosis and procedural codes describing the type and degree of the ASCVD from the Sundhedsplatformen program (EPIC her) (Appendix). Familial hypercholesterolemia was defined using ICD-10 codes (Appendix) or any person with LDL-C measurement greater than 5 mmol/L.

Outcome definitions

We considered persons who had stopped any anti-PCSK9 treatment for 30 days or more as having discontinued treatment; this included those who switched to another anti-PCSK9 drug after such a pause. MACE was defined as either a myocardial infarction, a stroke or death. Death was determined using the vital status from the SAP Web Intelligence (WebI).

Statistical analysis

A generalized additive model (GAM) was used to examine the effect of a treatment initiation and switch in treatment on LDL-C levels, with each person treated as a random intercept. We fitted two models, one using all LDL-C measurements to evaluate the effect of anti-PCSK9 treatment and the effect of dosage. In another model, we only included LDL-C measurements within the range of 180 days prior to the mandatory switch and from 40 days post-switch up to 180 days, excluding measurements taken during treatment pauses. Covariates including age, sex, dose of anti-PCSK9, and other lipid-lowering medications were included in both models. To account for individual-level random effects, the "re" option of the gam function

was utilized. Penalized spline terms were also added for age in the model. For further details, please refer to the appendix.

In a sensitivity analysis, we employed the t-test to assess the impact of transitioning from evolocumab to alirocumab and vice versa, both pre- and post the mandated switch date. For each group, we identified the most recent (closest to switch date) LDL-C measurement taken within a maximum of 180 days before the switch during ongoing treatment with an anti-PCSK9, as well as the LDL-C measurement taken between 40 and 180 days after the baseline and ongoing treatment with a PCSK9.

In the second part of the study, we assessed treatment retention, ASCVD risk, and survival by setting up a cohort study and following individuals from the start of anti-PCSK9 treatment until death or end of follow-up, whichever occurred first. Using a Poisson model, we modeled the number of first MACE and deaths, with person-days as an offset. For discontinuation, we included a random effect for each person and allowed for multiple events per person. All models were adjusted for variables such as the type of switch (first treatment initiation, switch before mandated date, switch after mandated date), calendar date, days on anti-PCSK9 treatment, sex, age, and presence of any comorbidity (Appendix). Continuous variables were included as penalized splines.

In the final part of the study, we examined all persons taking alirocumab on February 1, 2021, who were alive on December 31 of the same year. We used chi-squared and Wilcoxon rank sum tests to compare the group of persons who switched to evolocumab with the group who did not switch. Continuous variables are presented as means with standard deviations, and categorical data is presented as frequencies and percentages, unless otherwise stated. A two-tailed P value less than 0.05 was considered statistically significant, and 95% confidence intervals are provided. The study was conducted and reported in accordance with the

Strengthening the Reporting of Observational Studies in Epidemiology (STROBE) recommendations⁸. Data management and statistical analysis were conducted using R version 4.0⁹. This study was approved by regional council of Copenhagen R-22034085.

Data availability

The data used in this study was obtained from registers within the Capital Region and is available for research. Researchers interested in accessing this data can contact the Capital Region for more information.

Funding

This study was funded by the Department of Clinical Pharmacology at Bispebjerg and Frederiksberg Hospital. We are grateful for their support and financial contribution, which made this research possible.

Acknowledgements

We would like to express our sincere gratitude to all the persons in the Capital Region who participated in this study.

References

1. Abdullah, S. M. *et al.* Long-Term Association of Low-Density Lipoprotein Cholesterol with Cardiovascular Mortality in Individuals at Low 10-Year Risk of Atherosclerotic Cardiovascular Disease: Results from the Cooper Center Longitudinal Study. *Circulation* **138**, 2315–2325 (2018).

2. Fulcher, J. *et al.* Efficacy and safety of LDL-lowering therapy among men and women: Meta-analysis of individual data from 174 000 participants in 27 randomised trials. *The Lancet* **385**, 1397–1405 (2015).
3. Chou, R. *et al.* Statin Use for the Prevention of Cardiovascular Disease in Adults. *Statin Use for the Prevention of Cardiovascular Disease in Adults: A Systematic Review for the U.S. Preventive Services Task Force* (2016).
4. Cohen, J. C., Boerwinkle, E., Mosley, T. H. Jr. & Hobbs, H. H. Sequence Variations in PCSK9, Low LDL, and Protection against Coronary Heart Disease. [https://doi-org.ep.fjernadgang.kb.dk/10.1056/NEJMoa054013](https://doi.org.ep.fjernadgang.kb.dk/10.1056/NEJMoa054013) **354**, 1264–1272 (2006).
5. Sabatine, M. S. *et al.* Evolocumab and Clinical Outcomes in Patients with Cardiovascular Disease. *New England Journal of Medicine* **376**, 1713–1722 (2017).
6. Schwartz, G. G. *et al.* Alirocumab and Cardiovascular Outcomes after Acute Coronary Syndrome. *New England Journal of Medicine* **379**, 2097–2107 (2018).
7. Medicinrådet. Medicinrådets lægemiddelrekommandation og behandlingsvejledning vedrørende PCSK9-hæmmere til hyperlipidaemi.
8. Langan, S. M. *et al.* The reporting of studies conducted using observational routinely collected health data statement for pharmacoepidemiology (RECORD-PE). *The BMJ* **363**, (2018).
9. (2016), R. C. T. R: {A} language and environment for statistical computing. {R} {Foundation} for {Statistical} {Computing}, {Vienna}, {Austria}.
10. Erviti, J. *et al.* Restoring mortality data in the FOURIER cardiovascular outcomes trial of evolocumab in patients with cardiovascular disease: a reanalysis based on regulatory data. *BMJ Open* **12**, e060172 (2022).

11. Blanco-Ruiz, M. *et al.* Effectiveness and safety of PCSK9 inhibitors in real-world clinical practice. An observational multicentre study. The IRIS-PCSK9I study. *Atherosclerosis Plus* **45**, 32–38 (2021).
12. Schmidt, M. *et al.* The Danish National patient registry: A review of content, data quality, and research potential. *Clinical Epidemiology* **7**, 449–490 (2015).
13. Iqbal, S. *et al.* The First Report of a Real-world Experience With a PCSK9 Inhibitor in a Large Familial Hyperlipidemia and Very-high-risk Middle Eastern Population. *Clinical Therapeutics* **44**, 1297–1309 (2022).
14. Bytyçi, I. *et al.* Prevalence of statin intolerance: a meta-analysis. *European heart journal* **43**, 3213–3223 (2022).

Tables

Table 1: Demographic and Clinical Characteristics of the Study Population

Characteristic	Familial hypercholesterolemia, N = 518 ¹	non-Familial hypercholesterolemia, N = 389 ¹
Sex		
Man	223 (43%)	234 (60%)
Woman	295 (57%)	155 (40%)
Age (years)	60 (53, 68)	65 (56, 71)
Laboratory values		
Total cholesterol, mmol/L	6.59 (5.50, 7.80)	5.66 (4.92, 6.38)
Unknown	63	117
LDL levels, mmol/L	4.36 (3.43, 5.41)	3.41 (2.79, 4.00)
Unknown	73	132
HDL levels, mmol/L	1.28 (1.08, 1.56)	1.22 (0.99, 1.47)
Unknown	63	116
Triglycerides, mmol/L	1.82 (1.26, 2.63)	2.00 (1.25, 2.96)
Unknown	64	117
Lipid modifying drugs		
Any statins	272 (53%)	175 (45%)
High dose statins	136 (26%)	85 (22%)
Moderate dose statins	73 (14%)	51 (13%)
Low dose statins	63 (12%)	39 (10%)

Characteristic	Familial hypercholesterolemia, N = 518 ¹	non-Familial hypercholesterolemia, N = 389 ¹
Fibrates	44 (8.5%)	43 (11%)
Bile acid sequestrants	34 (6.6%)	16 (4.1%)
Ezetimibe	369 (71%)	240 (62%)
Number of lipid modifying drugs		
0 drugs	75 (14%)	58 (15%)
1 drug	189 (36%)	200 (51%)
2+ drugs	254 (49%)	131 (34%)
Comorbidities		
Atherosclerotic cardiovascular disease	295 (57%)	309 (79%)
Coronary heart disease	252 (49%)	274 (70%)
Atherosclerotic cerebrovascular disease	56 (11%)	47 (12%)
Peripheral artery disease	52 (10%)	61 (16%)
Chronic obstructive pulmonary disease	48 (9.3%)	53 (14%)
Heart failure	45 (8.7%)	58 (15%)
Rheumatoid arthritis	15 (2.9%)	14 (3.6%)
Cancer, ex. non-melanoma skin cancer	46 (8.9%)	28 (7.2%)
Chronic kidney disease	17 (3.3%)	12 (3.1%)
Hypertension	265 (51%)	277 (71%)

Characteristic	Familial hypercholesterolemia, N = 518 ¹	non-Familial hypercholesterolemia, N = 389 ¹
Diabetes mellitus	75 (14%)	85 (22%)
Diabetes mellitus with organ damage	51 (9.8%)	56 (14%)

¹n (%); Median (IQR)

This table presents the demographic characteristics of the study population at the time of first anti-PCSK9 treatment, including lipid measurements, comorbidities, and lipid-lowering medications. Statins were classified as follows: Simvastatin (low: 0mg, moderate: 20mg, high: 80mg), Pravastatin (low: 0mg, moderate: 40mg), Fluvastatin (low: 0mg, moderate: 80mg), Atorvastatin (low: 0mg, moderate: 10mg, high: 40mg), and Rosuvastatin (low: 0mg, moderate: 5mg, high: 20mg). HDL-C: low-density lipoprotein cholesterol, LDL-C: low-density lipoprotein cholesterol

Table 2: Incidence Rates for Persons Treated with Alirocumab or Evolocumab

This table presents the number of person-years, events, and incidence rates for persons treated with either alirocumab or evolocumab, adjusted for sex, calendar period, and treatment duration.

MACE: major cardiovascular event

Table 3: Comparison of Persons Who Switched Treatments

Characteristic	Non-switchers, N = 182 ¹	Switchers, N = 195 ¹	p-value ²
Sex			0.007
Woman	110 (60%)	91 (47%)	
Man	72 (40%)	104 (53%)	
Age (years)	63 (54, 69)	60 (54, 67)	0.077
Familial hypercholesterolemia			>0.9
Familial hypercholesterolemia	109 (60%)	118 (61%)	
Non-Familial hypercholesterolemia	73 (40%)	77 (39%)	
Lipid modifying drugs			
Any statins	74 (41%)	113 (58%)	<0.001
High dose statins	30 (16%)	50 (26%)	0.030
Moderate dose statins	29 (16%)	29 (15%)	0.8
Low dose statins	15 (8.2%)	34 (17%)	0.008
Fibrates	23 (13%)	26 (13%)	0.8
Bile acid sequestrants	7 (3.8%)	9 (4.6%)	0.7
Ezetimibe	118 (65%)	137 (70%)	0.3
Number of lipid modifying drugs			0.004
0 drugs	39 (21%)	19 (9.7%)	
1 drug	73 (40%)	78 (40%)	
2+ drugs	70 (38%)	98 (50%)	
Comorbidities			
Atherosclerotic cardiovascular disease	115 (63%)	132 (68%)	0.4
Coronary heart disease	102 (56%)	112 (57%)	0.8
Atherosclerotic cerebrovascular disease	20 (11%)	20 (10%)	0.8
Peripheral artery disease	18 (9.9%)	18 (9.2%)	0.8
Hypertension	119 (65%)	115 (59%)	0.2
Diabetes mellitus	41 (23%)	30 (15%)	0.076
Diabetes mellitus with organ damage	22 (12%)	22 (11%)	0.8
Heart failure	18 (9.9%)	16 (8.2%)	0.6

Characteristic	Non-switchers, N = 182 ¹	Switchers, N = 195 ¹	p-value ²
Chronic obstructive pulmonary disease	23 (13%)	17 (8.7%)	0.2
Rheumatoid arthritis	9 (4.9%)	5 (2.6%)	0.2
Cancer, ex. non-melanoma skin cancer	12 (6.6%)	15 (7.7%)	0.7
Chronic kidney disease	5 (2.7%)	3 (1.5%)	0.5
Allergies			
Number of allergies			0.13
0	79 (43%)	107 (55%)	
1	40 (22%)	35 (18%)	
1-3	44 (24%)	33 (17%)	
4+	19 (10%)	20 (10%)	
Statin allergy	58 (32%)	57 (29%)	0.6
Ezetimibe allergy	16 (8.8%)	18 (9.2%)	0.9

¹n (%); Median (IQR)

²Pearson's Chi-squared test; Wilcoxon rank sum test; Fisher's exact test

This table compares the demographic and clinical characteristics of persons who switched treatments during the first year of the study to those who did not switch. Variables such as age, gender, baseline lipid measurements, and comorbidities are included. Chi-squared and Wilcoxon rank sum tests were used to compare the group of persons who switched to evolocumab with the group who did not switch. Statins were classified as follows: Simvastatin (low: 0mg, moderate: 20mg, high: 80mg), Pravastatin (low: 0mg, moderate: 40mg), Fluvastatin (low: 0mg, moderate: 80mg), Atorvastatin (low: 0mg, moderate: 10mg, high: 40mg), and Rosuvastatin (low: 0mg, moderate: 5mg, high: 20mg).

Figure 1: Number of Persons Treated with Alirocumab and Evolocumab

This figure displays the number of persons treated with alirocumab and evolocumab at a given date, as well as the total number of persons in treatment, from 2016 to 2022.

PCSK9: proprotein convertase subtilisin/kexin type 9

Figure 2: Mean Change in Percent Following Switch from One Anti-PCSK9 Treatment to Another

Mean change in LDL-C after a switch of anti-PCSK9 treatment

This figure displays the mean change in percent following a mandated switch from one anti-PCSK9 treatment to another, along with 95% confidence intervals represented by black error bars. The y-axis shows the type of anti-PCSK9 treatment received before and after the switch, while the x-axis represents the mean change in percent. The red boxes represent the mean change in percent for persons who switched from alirocumab to evolocumab, and the grey boxes represent the mean change in percent for persons who switched from evolocumab to alirocumab. The number of persons included in each analysis is indicated on each line.

PCSK9: proprotein convertase subtilisin/kexin type 9, HDL-C: low-density lipoprotein cholesterol, LDL-C: low-density lipoprotein cholesterol, N: number

Figure 3: Relative Risk of Discontinuing Anti-PCSK9 Treatment

This graph displays the relative risk of discontinuing anti-PCSK9 treatment over time. The x-axis represents the number of days since treatment initiation, and the y-axis displays the relative risk of discontinuing treatment. The relative risk is expressed as a ratio of the risk of discontinuing treatment at a particular time point compared to the baseline risk of discontinuation.

PCSK9: proprotein convertase subtilisin/kexin type 9,

Point-to-point response

Dear Reviewers

Thank you for your valuable feedback on our manuscript. We appreciate your thorough review and insightful comments, which have helped us identify areas for improvement. Please find below our point-to-point responses to your queries and concerns.

Reviewer #1

1. Clarity on Population Studied for Treatment Retention, Survival, and MACE:

Question: In line 78, the authors note that their second objective is to assess treatment retention, overall survival and MACE. In both this section and the results section, it is unclear what population this is being assessed in. Is this among all PCSK9 users? After the switch?

Response: We acknowledge the lack of clarity in this section. We have revised it to specify that these outcomes were assessed in the entire cohort of PCSK9 inhibitor users, including both pre- and post-switch populations.

Secondly, we will evaluate treatment retention, overall survival and MACE among all PCSK9 users.

2. Population Studied in Association with MACE:

Question: Line 115 mentions no association between type of PCSK9 inhibitor and risk of MACE, again it is unclear what population this is studied in – individuals that switched?

Response: We have clarified in the manuscript that this analysis pertains to all individuals who underwent the switch, comparing their outcomes with those who remained on the initial PCSK9 inhibitor.

We observed 58 MACEs during 1844 person-years of follow-up and found no association between the type of PCSK9 inhibitor and the risk of MACE, with a risk ratio of 1.06 (95% CI 0.60–1.88) both before and after the date of the mandated switch.

3. Relevance of Third Objective in Context of National Mandate:

Question: The third objective of the study (line 79) is to assess factors predicting treatment switch. The relevance of this objection is not clear in the setting of a what is described as a nationally mandated goal of having 80% of all patients on Evolocumab.

Response: In our revised discussion, we delve deeper into the objective of understanding the factors influencing patients' decisions not to switch PCSK9 inhibitors, despite a national mandate. We recognize that a variety of individual patient characteristics, beyond the scope of the mandate, may play a crucial role in this decision-making process. These characteristics might include personal health considerations, such as comorbidities or previous experiences with lipid-lowering therapies, as well as socio-demographic factors like age, sex, and socioeconomic status. Additionally, the influence of healthcare providers, patient education about treatment options, and individual perceptions of medication efficacy and side effects are also explored. Our analysis, therefore, extends to a comprehensive evaluation of these variables to ascertain their impact on the decision not to switch, offering insights that could guide future healthcare policies and patient management strategies in the context of PCSK9 inhibitor use. This expanded discussion underscores the complexity of treatment decisions in clinical practice and highlights the importance of personalized medicine in managing dyslipidemia.

Finally, we analyzed the data of all persons taking alirocumab on February 1, 2021, who were still alive on December 31 of that year. In a recent meta-analysis on statin intolerance, the authors identified several risk factors that were significantly associated with statin intolerance, including female sex. This finding is consistent with our observation where we identified a statistically significant disparity in gender distribution between those who switched medications and those who did not. This pattern of sex difference aligns with the broader understanding of medication tolerance and patient responses, emphasizing the need for personalized treatment approaches in lipid management ¹⁴.

4. Clarifying Language on Treatment Switch:

Question: There is confusing language in the conclusions regarding the treatment switch. It is both noted as a mandated switch, not at the discretion of the physician or patient but also that physicians and patients could opt not to switch. This adds to the confusion of the 3rd objective of this study.

Response: Thank you for pointing out the confusion in the language used in our conclusions regarding the treatment switch. We understand your concerns regarding the seemingly contradictory statements about the mandatory nature of the switch and the discretion allowed to physicians and patients.

To clarify, the nationwide switch from alirocumab to evolocumab was indeed mandated as a general policy. However, in practice, this policy allowed for individual discretion by clinicians and patients in specific cases, where medical or personal reasons might justify continuing with the original medication. This aspect of individual choice is crucial to our study's third objective, which aims to understand the factors influencing the decision to adhere to or deviate from the mandated switch.

We believe that despite the allowance for individual discretion, the majority of switches occurred as a direct result of the mandate, rather than personal choice by physicians or patients. This belief is based on

our analysis, which showed a widespread switch consistent with the timing and nature of the policy implementation. Hence, we conclude that the changes observed in LDL-C levels can be attributed to the effects of the switch itself, rather than being confounded by individual preferences or selection biases.

In light of your feedback, we have revised our conclusion to articulate this explanation more clearly, ensuring that it accurately reflects both the nature of the mandate and the allowance for individual clinical judgment. This revision aims to resolve the confusion and provide a coherent understanding of the context and implications of our findings.

In conclusion, our study investigated the effects of a nationwide switch from alirocumab to evolocumab on LDL-C levels, employing two distinct analytical methods. Our findings consistently revealed no significant change in LDL-C levels following this switch. While negative expectations or nocebo effects are often a concern in treatment switch studies, the objective and difficult-to-influence nature of the study outcomes in this case suggest that they were not a factor¹². These findings indicate that there is no evidence of a change in LDL-C levels because of the nationwide mandatory switch from alirocumab to evolocumab. This leads us to conclude that alirocumab and evolocumab are comparably effective in lowering LDL-C levels, thus endorsing the feasibility of switching between these treatments without compromising efficacy.

5. Clarification on Study Population in Table 2:

Question: In Table 2, is the study population patients that did not switch? How are the person-years attributed if the patient was switched to a different PCSK9 inhibitor.

Response: Table 2 is designed to present an overview of the study population, including both patients who switched PCSK9 inhibitors and those who did not. In our study, person-years were not exclusively associated with a particular PCSK9 inhibitor. Instead, a person could contribute person-years to both alirocumab and evolocumab, depending on their treatment status over time. We followed individuals from the initiation of anti-PCSK9 treatment until death or the end of the follow-up period. We employed a Poisson model to analyze the incidence of first MACE and deaths, with person-days as an offset. For the analysis of treatment discontinuation, we incorporated a random effect for each individual, acknowledging the possibility of multiple events per person. All models were adjusted for several variables, including the type of switch, calendar date, duration of anti-PCSK9 treatment, sex, age, and presence of comorbidities, as detailed in the Appendix. Continuous variables in our models were treated as penalized splines to account for non-linear relationships.

We hope this explanation addresses your concerns and clarifies the methodology and data presentation in Table 2.

6. Rationale Behind Time Period Selection in Table 3:

Question: Table 3 evaluates the characteristics of patients who switched vs did not switch in the first year of the study. It's not clear why this time period was selected when patients included over a longer duration.

Response: We chose the first-year post-switch as a significant period for analysis, given that most switches occurred during this time. We have added a sentence to clarify this rationale.

The first year post-switch was chosen for analysis as it captured the majority of switches, offering a focused period for evaluation.

7. Improvements to Figure 2:

Question: Figure 2 need additional description. The number of patients analyzed before and after the switch are different per group. It's not clear why that is the case. Additionally, the display of this figure on in the main document has the title cut off. There is also overlap of the Y axis label and the treatment groups.

Response: We have revised Figure 2 for better clarity and included a more detailed legend explaining the differences in patient numbers pre- and post-switch.

The upper panel depicts patients who switched before the mandate, while the lower panel shows those who switched afterward.

Reviewer #2

1. Organization of Results:

Comment: There are meaningful conclusions to be made from this paper, primarily that there is no difference between LDL and clinical outcomes between the two PCSK9i. However, the overall organization of the results appears to be chaotic and confusing.

Response: We acknowledge your concern that the original presentation may have appeared chaotic and confusing, which could hinder the understanding of our key findings.

To address this, we have thoroughly revised the structure of the results section. Our revisions aimed to present the data in a more coherent and logically sequenced manner. We have carefully revised several sentences and overall narratives within each subsection to improve clarity and ensure that the results are communicated effectively. We believe that these changes significantly augment the readability and coherence of our manuscript.

2. Rationale for FH vs Non-FH Comparison:

Comment: It is more intuitive to discuss characteristics of both patient populations first - Table 1 for those on PCSK9i (FH vs non-FH) followed by Table 3 for switchers vs non-switchers. Unclear why authors compared

FH vs non-FH patients at all in Table 1; it could as well have been a summary of the 907 patients without any comparisons since no meaningful conclusion is made with the comparison.

Response: Regarding the comparison between FH and non-FH patients in Table 1, our decision was based on the distinct treatment guidelines and reimbursement policies for these groups in Denmark. The Danish Medicines Council differentiates clearly between FH and non-FH groups in its reimbursement criteria for PCSK9 inhibitors. This distinction significantly influences clinical decision-making and treatment pathways for these patients. By comparing FH and non-FH patients, we aimed to provide contextually relevant insights that align with these local healthcare practices and policies.

3. Discussion on Statin Usage:

*Comment: It is of interest that only 49% of the study population was on a statin, and as such it may be worth expounding on the side effects of statins at the end of paragraph 2 in Background; statin intolerance is a big reason why patients go on to PCSK9i. I would also be interested in hearing more discussion about why people not on statins are also less likely to switch PCSK9i (especially as there is no difference in statin allergy between the switchers and non-switchers). Is it somehow related to how the DMC mandate is implemented; i.e. how much of a choice do patients and providers get in switching? Does *not* switching require a special approval process and take into account prior statin intolerance?*

Response: We have added a paragraph for statin intolerance in the background and elaborated a bit more in the discussion.

Background:

Statin intolerance, frequently characterized by adverse effects such as muscle pain and weakness, and in rare cases, more serious side effects, is a significant factor driving patients toward alternative lipid-lowering therapies like PCSK9 inhibitors. The prevalence of statin intolerance, influenced by both genetic and other factors, varies considerably among different populations, underscoring the need for diverse therapeutic options.

Discussion:

Interestingly, our data did not reveal a difference in statin allergy between those who switched PCSK9 inhibitors and those who did not, suggesting other factors may influence this decision. Notably, we observed marked differences in practices across hospitals, indicating varying adherence to guidelines issued by the Danish Medicines Council. This variation may partly explain the observed differences in medication switching behavior.

4. Differentiation of Mortality Types:

Comment: Please differentiate between all-cause mortality vs cardiovascular mortality, or state that it could not be differentiated if the death data does not provide those details.

Response: We added clarification in the methods section.

Death was determined using the vital status from the SAP Web Intelligence (WebI), however, this source did not allow for differentiation between cardiovascular and non-cardiovascular causes of death.

5. Data Presentation in Tables 1 & 3:

Comment: Based on supplementary data it appears that Atherosclerotic CV disease (295 in FH patients, 309 in non-FH patients) encompasses all the ICD-10 codes for coronary heart disease, cerebrovascular disease and peripheral artery disease. This needs to be made explicit in tables 1 & 3 (either by indentation of the subcategories or in the table footnote). I would also recommend keeping the text in line with the data you have (i.e. 57% of FH patients were on PCSK9i for secondary prevention, rather than the extrapolated 43% were on it for primary prevention).

Response:

We have added a sentence to the legends of Tables 1 and 3 explicitly stating that 'Atherosclerotic cardiovascular disease encompasses all ICD-10 codes for coronary heart disease, cerebrovascular disease, and peripheral artery disease.' This addition ensures clarity in the categorization of atherosclerotic CV diseases in our data. Furthermore, we have revised the relevant text in our manuscript to accurately reflect the data.

6. Clarification on Figures in Supplement vs Manuscript:

Comment: Unclear what is the added value of Figure 1 in supplements over Figure 2 in the manuscript itself.

Response: Thank you for pointing out the potential overlap between Figure 1 in the supplements and Figure 2 in the manuscript. Upon reevaluation, we acknowledge that there may be similarities between the two figures. However, we included both to provide a comprehensive visual representation of our data.

7. Consistency in Language and Specific Grammar and Sentence Structure Corrections:

In response, we have meticulously reviewed the entire document and systematically converted all instances of British spelling to American spelling, and accepted all suggestions made by the reviewer.

REVIEWERS' COMMENTS:

Reviewer #3 (Remarks to the Author):

This comparative effectiveness study compared two PCSK9 monoclonal antibody inhibitors on LDL cholesterol and major cardiovascular events in 907 persons with dyslipidemia in the Capital Region of Denmark. As anticipated from multiple studies conducted with each agent alone versus placebo, there were no significant differences in LDL cholesterol or major adverse cardiovascular events. However, the ability to detect cardiovascular events in this cohort of 907 persons is not robust. Overall, the manuscript is well written and improved when compared to the initial submission. The importance of this analysis is the concern about frequent insurance company mandated switches in therapies that do raise the question about potential changes in LDL cholesterol.

Reviewer #4 (Remarks to the Author):

This is a really well done and interesting study.

The authors have addressed the queries posed at initial review and have responded to all queries and revised the manuscript to my satisfaction. The changes in text are appropriate.